# Cutaneous Melanomas: A Single Center Experience on the Usage of Immunohistochemistry Applied for the Diagnosis

**DOI:** 10.3390/ijms23115911

**Published:** 2022-05-25

**Authors:** Costantino Ricci, Emi Dika, Francesca Ambrosi, Martina Lambertini, Giulia Veronesi, Corti Barbara

**Affiliations:** 1Pathology Unit, Ospedale Maggiore, 40139 Bologna, Italy; costanricci@gmail.com (C.R.); fra.ambrosi@gmail.com (F.A.); 2Department of Experimental, Diagnostic and Specialty Medicine (DIMES), University of Bologna, 40139 Bologna, Italy; emi.dika3@unibo.it; 3Dermatology Unit, IRCCS Policlinico Sant’Orsola-Malpighi, University of Bologna, 40139 Bologna, Italy; martinalambertini@gmail.com (M.L.); giulia.veronesi.md@gmail.com (G.V.); 4Pathology Unit, IRCCS Azienda Ospedaliero-Universitaria di Bologna, Policlinico di Sant’Orsola, 40139 Bologna, Italy

**Keywords:** cutaneous melanoma, skin melanoma, melanoma, immunohistochemistry, immunohistochemical markers, diagnosis

## Abstract

Cutaneous melanoma (cM) is the deadliest of all primary skin cancers. Its prognosis is strongly influenced by the stage at diagnosis, with early stages having a good prognosis and being potentially treatable with surgery alone; advanced stages display a much worse prognosis, with a high rate of recurrence and metastasis. For this reason, the accurate and early diagnosis of cM is crucial—misdiagnosis may have extremely dangerous consequences for the patient and drastically reduce their chances of survival. Although the histological exam remains the “gold standard” for the diagnosis of cM, a continuously increasing number of immunohistochemical markers that could help in diagnosis, prognostic characterization, and appropriate therapeutical choices are identified every day, with some of them becoming part of routine practice. This review aims to discuss and summarize all the data related to the immunohistochemical analyses that are potentially useful for the diagnosis of cM, thus rendering it easier to appropriately applicate to routine practice. We will discuss these topics, as well as the role of these molecules in the biology of cM and potential impact on diagnosis and treatment, integrating the literature data with the experience of our surgical pathology department.

## 1. Introduction

Cutaneous melanoma (cM) is a malignant and potentially lethal tumor that develops from the transformation of the melanocytes that normally reside in the basal layer of the skin epidermis and form with the keratinocytes the epidermal melanin unit [1,2]. The annual incidence and morbidity of cM are constantly increasing worldwide (the number of newly diagnosed cases has more than doubled since 1973), probably due to population aging, the increase of risk factors such as chronic sun damage, and the improvement of diagnostic tools; besides, unlike other malignancies, cM affects a higher proportion of younger patients (median age: 57 years), with a sex preponderance that varies in different age groups (female preponderance in younger age groups (4:10 in 20–30-year-olds); male preponderance (16:10 in >85-year-olds)) [3,4]. cM is also the most lethal cutaneous tumor, with mortality rates ranging between 3.5/100,000 in Australia and 1.7/100,000 in Europe [3,4]. The appropriate and early diagnosis of cM is crucial for improving the prognosis and therapeutic strategies of the affected patients [3,4]. In this scenario, the immunohistochemistry, as an essential and indispensable aid to the correct histological diagnosis, plays a key role [1,2,3,4]. This review aims to present all the data related to the immunohistochemistry of cM, discussing their application for diagnosis, prognostic characterization, and treatment of this deadly disease, as well as briefly summarizing the role of these molecules in the biology of melanocytes and cM.

## 2. Diagnosis

### 2.1. Histological Exam

Despite an everyday increasing understanding of molecular biology and the etiology of cM, the diagnosis of cM is mainly performed by pathologists, with the histological exam rendered on hematoxylin and eosin (H&E) slides [5,6]. The differential diagnosis between cM and cutaneous nevus (cN) is based on the identification and assessment of numerous morphological criteria. Nevertheless, no criterion are completely specific for cM, and all of them could also potentially be found in cN; some criterion are found only in specific cN and cM subtypes, and others are qualitatively assessed and suffer from low interobserver agreement [5,6]. Besides, new histological entities of cN and cM are identified every day based on different clinical–pathological and molecular backgrounds [5,6,7,8]. As result, the diagnosis of cM remains one of the most difficult challenges of surgical pathology, and it should be rendered by only dedicated dermatopathologists that integrate the histological exam rendered on H&E slides, with available clinical, immunohistochemical, and molecular data [5,6,9,10]. In this scenario, the immunohistochemistry is certainly the most largely adopted tool for supporting the histological diagnosis of melanocytic lesions [5,6,9,10].

### 2.2. Immunohistochemistry

Despite the continuous development of molecular genetic diagnostic techniques, immunohistochemistry remains the most frequently performed and cost-effective tool for implementing the histological exam for the diagnosis of cM. In this review, we analyzed the immunohistochemical markers that were preferentially adopted by us and other surgical pathology laboratories for the diagnosis of melanocytic lesions, along with their expression profile, routinary use and clones, potential diagnostic pitfalls, and ongoing research topics. Furthermore, for each of these markers, we will briefly summarize the role performed in the biology and embryology of the melanocytes, as well as in the genesis of cN and cM. For practical purposes, we divided them into four major classes (in *italic*, we reported the markers subsequently described):

Melanocytic differentiation markers (*S100*, *SOX10*, *HMB-45*, *Melan A*/*MART-1*, MITF, Tyrosinase, KBA 6.2, NKI/beteb, PNL2, MC1R, CD146/Mel-CAM, NKI/C3, and p75NGFR).

Markers useful for the differential diagnosis between CN and CM (*HMB-45*, *Ki67*, *p16*, *p21*, *p53*, *PRAME*, NKI/beteb, 5-hmC, PTEN, PHH3, H3KT, and H3KS).

Markers useful for the identification of specific histological subtypes of CN and CM (*BRAF V600E*, *c-Kit/CD117*, *ROS1*, *ALK*, *pan-TRK, BAP-1*, *β-catenin*, *PRKAR1A*, *NF1*, and *IDH1*).

Double stains (DS) (*HMB-45/Ki67*, *MART-1/Ki67*, D2-40/MITF, D2-40/S-100, D2-40/SOX10, D2-40/MART-1, *CD34/SOX10*, *HMB-45/PRAME*, *MART-1/PRAME*, and MART-1/PHH3).

Some of these markers could belong to more than one class (HMB-45) and have been discussed only in one of them. A summary of the main application fields of the immunohistochemical markers that are most frequently adopted for the diagnosis of cM is presented in Table 1. Illustrative examples of the immunohistochemical markers that are adopted in complex routine diagnostic cases are shown in Figure 1 and Figure 2.

**Table 1 ijms-23-05911-t001:** Summary of the main application fields for the immunohistochemical markers that are most frequently adopted for the diagnosis of cM.

Immunohistochemical Markers	Main Application Fields for the Diagnosis of cM
S100	(1) diagnosis of metastasis of unknown primary tumor;
	(2) diagnosis of primary cutaneous tumor with undifferentiated morphology;
	(3) diagnosis of desmoplastic cM;
	(4) identification of MM and NN in SLNB;
SOX10	(1) diagnosis of metastasis of unknown primary tumor;
	(2) diagnosis of primary cutaneous tumor with undifferentiated morphology;
	(3) diagnosis of desmoplastic cM;
	(4) identification of MM and NN in SLNB;
	(5) assessment of the nuclear profile of melanocytes (useful for the grading of melanocytic dysplasia in dysplastic cN);
	(6) correct estimation of the spread of lentiginous melanocytic proliferations;
	(7) correct assessment of the depth of invasion (Breslow thickness);
	(8) identification of the lympho-vascular invasion, adnexal involvement, and peri-adnexal extension in cM;
	(9) correct estimation of the intra-epithelial pagetoid spreading;
	(10) differential diagnosis between scar and desmoplastic cM (especially in the excisional enlargement of desmoplastic cM);
HMB-45	(1) diagnosis of metastasis of unknown primary tumor;
	(2) diagnosis of primary cutaneous tumor with undifferentiated morphology;
	(3) identification of MM (and differential diagnosis with NN) in SLNB;
	(4) evaluation of the gradient of melanocytic maturation in cN (present) and cM (absent and/or altered);
	(5) evaluation of the junctional component of cN and cM (useful for the grading of melanocytic dysplasia in dysplastic cN);
	(6) distinction between the dermal component of cN and cM (mainly nevoid cM);
Melan A/MART-1	(1) diagnosis of metastasis of unknown primary tumor;
	(2) diagnosis of primary cutaneous tumor with undifferentiated morphology;
	(3) identification of MM and NN in SLNB;
	(4) evaluation of the silhouette (symmetry/asymmetry) of cN and cM;
	(5) correct estimation of the depth of invasion in cM;
	(6) identification of the lympho-vascular invasion, adnexal involvement, and peri-adnexal extension in cM;
Ki67	(1) evaluation of the proliferation index (absolute value);
	(2) evaluation of the “dermal hot-spot” </≥5%, unusual/deep/asymmetrical staining pattern of the dermal component, Ki67(+) deep dermal cells with pleomorphism atypical nuclei, and Ki67(+) intraepithelial cells exhibiting pagetoid spreading;
p16	(1) evaluation of dermal and/or nodular atypical melanocytic lesions/melanocytomas;
	(2) identification of a more aggressive phenotype acquired by the primary cM;
	(3) identification of MM (and differential diagnosis with NN) in SLNB;
p21	(1) evaluation of Spitz melanocytic lesions (especially acral);
	(2) evaluation of mucosal melanocytic lesions;
p53	(1) differential diagnosis between neurofibroma-like desmoplastic cM and neurofibroma;
PRAME	(1) evaluation of ambiguous melanocytic lesions (able to distinguish cM (PRAME+) from cN (PRAME-), with a high concordance rate via cytogenetic tests);
	(2) differential diagnosis between NN and MM in selected difficult cases;
	(3) evaluation of surgical resection margins in lentigo maligna;
	(4) distinction between the dermal “nevoid” component of nevoid cM and dermal cN in nevus-associated cM;
	*The introduction of PRAME for the diagnosis of melanocytic pathology is recent, and the fields of potential application are continuously evolving, as well as technical issues (cut-offs, interpretation of intermediate values, and discordant cases with the molecular tests);*
BRAF V600E, c-Kit/CD117, ALK, ROS1, pan-TRK (NTRK1, NTRK2, NTRK3), RET, MET, β-catenin, PRKAR1A, BAP-1, NF1, and IDH1	(1) identification of specific histological entities, characterized by specific molecular alterations (also see Table 2);
	(2) identification of potential therapeutic targets and increase of therapeutic choices;
HMB-45/Ki67 and MART-1/Ki67	(1) correct assessment of Ki67 index in melanocytic lesions, almost exclusively junctional/intraepithelial;
	(2) correct assessment of Ki67 index in melanocytic lesions with a high inflammatory infiltrate;
CD34/SOX10	(1) Identification of the lympho-vascular invasion in cM;
HMB-45/PRAME and MART-1/PRAME	(1) correct assessment of PRAME score in melanocytic lesions, almost exclusively junctional/intraepithelial;
	(2) correct assessment of PRAME score in melanocytic lesions with a high inflammatory infiltrate;
	(3) differential diagnosis between NN and MM in selected difficult cases;
	(4) diagnosis of metastasis of unknown primary tumor (especially with limited available histological material);
	(5) diagnosis of primary cutaneous tumor with undifferentiated morphology (especially with limited available histological material);

Abbreviations: cM (cutaneous melanoma); cN (cutaneous nevus); NN (nodal nevus); MM (melanoma metastasis); SLNB (sentinel lymph node biopsy).

**Table 2 ijms-23-05911-t002:** Summary of the immunohistochemical markers useful for the identification of specific histological subtypes of cN and cM.

Immunohistochemical Markers	Histological Entities Related to Their Over- and/or Aberrant Expression
BRAF V600E	(1) melanocytic lesions in intermittently sun-exposed skin (superficial spreading cM, simple lentigo, conventional and/or lentiginous cN, and dysplastic cN);
	(2) deep-penetrating cN (together with β-catenin);
	(3) *BAP1*-inactivated melanocytic lesions (together with BAP1);
	(4) PEM (together with PRKAR1A);
	(5) metastatic cM;
	(6) more rarely other melanocytic lesions [naevoid cM, nodular cM, and acral melanocytic lesions (especially cM), etc.];
c-Kit/CD117	(1) acral melanocytic lesions (especially cM);
	(2) lentigo maligna;
ALK, ROS1, TRK (NTRK1, NTRK2, NTRK3; all of them identified by immunohistochemistry for pan-TRK), RET and MET	(1) Spitz lesions (Spitz nevus, atypical Spitz tumor, and Spitz melanoma), including Reed cN;
	(2) acral melanocytic lesions (especially cM);
	(3) more rarely other melanocytic lesions (nodular cM, superficial spreading cM, etc.);
PRKAR1A	(1) PEM (together with BRAF V600E);
BAP1	(1) *BAP1*-inactivated melanocytic lesions (together with BRAF V600E);
	(2) cM arising in blue cN and atypical cellular blue tumor;
β-catenin	(1) deep-penetrating cN (together with BRAF V600E);
	(2) rare cases of cM with a “deep-penetrating like silhouette”;
NF1	(1) lentigo maligna;
	(2) desmoplastic cM;
	(3) acral melanocytic lesions (especially cM);
IDH1	recently introduced category of melanocytoma;

cM (cutaneous melanoma); cN (cutaneous nevus); PEM (pigmented epithelioid melanocytoma).

#### 2.2.1. Melanocytic Differentiation Markers

##### S100

The S100 protein family comprises about 25 members that are encoded by different genes located on chromosome 1q21 (in the so-called epidermal differentiation cluster, an area frequently rearranged in several tumors) and have the ability to form homodimers, heterodimers, and oligodimers [11,12,13,14]. Members of the S100 protein family are multifunctional proteins that, through the interaction with various effectors, are involved in a wide variety of cellular processes (cell growth, cell cycle regulation, protein secretion, contraction, cytoskeleton arrangement, etc.) [11,12,13,14]. Alteration in the levels of S100 proteins have been detected in several conditions (tumors, inflammatory disorders, genetic disease, and neurodegenerative conditions), as well as specific subtypes that could be differently implicated in tumor progression and suppression [11,12,13,14]. The most commonly used antibodies against S100 in routine practice are mouse and rabbit monoclonal antibodies (clones SHB1, 9A11B9, and SP127 (used in our laboratory)), and they are used directly against the S100B protein subtype [15,16]. S100 is probably the most historically known and commonly used melanocytic differentiation marker in surgical pathology laboratories, with it being expressed in almost all cN and cM (as well as desmoplastic cM) [17,18,19,20]. Its sensitivity ranges between 93% and 100% in the published series, with a characteristic staining pattern in both the nucleus and cell cytoplasm; however, S100 is not highly specific, with it also being expressed by several soft tissue tumors (nerve sheath tumors, adipocytic tumors, chondroid tumors, notochordal tumors, and many others), hematopoietic disorders (Langerhans cell histiocytosis), and others tumors (glial tumors, sex cord-stromal tumors, myoepithelial carcinoma, and other salivary gland tumors) [17,18,19,20,21,22,23,24,25]. For this reason, we always recommend using S100 in conjunction with other melanocytic (HMB-45 and MART-1) and case-by-case selected immunohistochemical markers, in specific diagnostic settings (metastasis of unknown primary cutaneous tumors with undifferentiated morphology). On the other hand, taking into account the high sensibility of S100, this marker has been largely used for the detection of melanoma metastases (MMs) in sentinel lymph node biopsy (SLNB) [26,27]. However, as S100 could label histiocytic and dendritic cells in lymph nodes, in the past years, we always added HMB-45 and have recently started to substitute it with SOX10.

##### SOX10

The sex determinant region Y-box 10 (SOX10) is a member of a family of approximately 20 transcription factors, and it is encoded by a gene located on chromosome 22q13.1 and specifically involved in the development of neural crest, peripheral nervous system, and melanocytes [28,29]. SOX-10 is expressed by the neural crest stem cells during embryonic development (dorsal neural tube) and later marks all the histological structures originating from the migration of these cells to specific embryonic sites [28,29]. SOX10 is a protein of 466 amino acids (about 58 kd), which are normally expressed by melanocytes, Schwann cells in peripheral nerves, oligodendrocytes, and myoepithelial cells, whose genetic mutations is associated with Waardenburg–Shah syndrome (hypopigmentation of the skin, heterochromia irides, deafness, and Hirschsprung disease) [28,29]. At present, several anti-SOX10 antibodies are commercially available, among which are the clones 1E6 (used in our laboratory) and A-2 [30,31,32,33,34,35]. SOX10 is universally accepted as the most sensitive marker for cN and cM (98–100%) in metastatic cM and 78–100% in desmoplastic cM), with the advantage of not staining dendritic and/or histiocytic cells in lymph nodes; as result, it is largely preferred to S100 for the evaluation of SLNB, with the updated EORTC protocol and characterization of unknown primary metastatic and/or primary cutaneous undifferentiated tumor [27,30,31,32,33]. However, similarly to S100, SOX10 exhibits a low specificity, being potentially expressed by a large number of tumors (carcinomas and soft tissue tumors), and it should be always used in conjunction with other immunohistochemical markers, depending on the diagnostic scenarios [32,34,35]. The staining pattern of SOX10 is nuclear and provides a cleaner signal, compared to cytoplasmatic (HMB-45 anfd MART-1) and cytoplasmatic/nuclear (S100) melanocytic markers; for this reason, in our personal experience, its results are more appropriate for highly pigmented lesions, as well as the evaluation of the nuclear profile (useful for the assignment of melanocytic dysplasia according to WHO 2018 criteria) and correct estimation of intra-epithelial pagetoid spreading. In our experience and according to previous papers, an additional advantage of SOX10 is its potential application for the differential diagnosis between proliferating fibroblasts of scar (SOX10(−)) and the residual component of desmoplastic cM (SOX10(+)) in excisional enlargements [36]. However, some authors found that SOX10 could be a potential source of dangerous diagnostic pitfalls (positive histiocytes, frozen section in Mohs surgery, entrapped nerves, and fibroblasts with “neuroid” features) and argue against its adoption in this scenario [37,38]. Based on our experience, we suggest always comparing SOX10 with the contextual H&E slides analysis to verify the morphology of (SOX10(+)) cells in the excisional enlargements of desmoplastic cM.

##### HMB-45

The name HMB-45 (human melanoma black) originated to indicate the immunogen associated with the monoclonal antibody and targeting PMEL17/gp100, which is a membrane-bound melanosomal protein encoded by a gene located on chromosome 12q13-q14; it is involved in the intracellular organization of melanosomes [39,40]. Electron microscopy studies showed that HMB-45 primarily reacts with the fibrillar matrix in stage II melanocytes (appearing as intraluminal striations in stage II melanosomes—above which, the melanin is deposited in stage III melanosomes), and it secondarily reacts with multivesicular stage I melanosomes [39,40]. The most frequently adopted antibody (also in our laboratory) to detect HMB-45 in routine practice is the monoclonal mouse antibody, clone HMB-45 [40]. HMB-45 has a lower sensibility as a melanocytic marker, compared to S-100 and SOX-10 (73–100% in primary cutaneous cM, 58–95% for metastatic cM and only 9–15% in desmoplastic cM), so the latter should be preferred for the immunohistochemical characterization of unknown primary metastatic and/or primary cutaneous undifferentiated tumors [40,41,42]. Nevertheless, HMB-45 is negative in most of the tumors that could histologically mimic cM and be positive for S-100 and SOX-10, so we often add it to the immunohistochemical panels adopted in these diagnostic settings [32,43]. HMB-45 could turn out positive in PEComa and related tumors, melanotic schwannoma, clear cell sarcoma, sex cord-stromal tumors, MiT family translocation renal cell carcinomas, pheochromocytoma, and rare cases of salivary gland tumors (as previously specified, it reacts with the fibrillar matrix in stage II melanocytes and should be more appropriately considered an organelle-specific marker, rather than a lineage-specific marker) [44,45,46,47]. In the melanocytic lesions, HMB-45 strongly reacts with the junctional and intraepidermal melanocytes; in our experience, it is the best marker for the evaluation of the junctional component, with an intensity that correlates with the grade of the dysplasia in dysplastic cN [48,49]. By contrast, the dermal component of cN is completely negative for HMB-45 and/or tends to retain it only in the superficial portion and loses it with maturation, differently from the dermal component of cM (mainly nevoid cM) that retains the stain (diffusely or patchy/focaln with isolated and/or clustered cells in both the superficial and deep parts of the lesion) [48,49]. However, dermatopathologists are aware that this axiom has several exceptions in routinary diagnostic practice: (1) blue cN, deep-penetrating cN, and other benign dermal melanocytosis are usually HMB-45(+); (2) nevoid cM could be completely HMB-45(-) in the dermal component exhibiting the so-called “pseudo-maturation” [48,49,50,51,52]. An additional diagnostic field for HMB-45 is the differential diagnosis between nodal nevi (NN) (HMB45(-)) and MM (HMB45(+)) in the pathological evaluation of SLNB [53]. Nevertheless, according to the literature data, as well as in our experience, p16 and PRAME (NN: p16(+) and PRAME(-); MM: p16(-) and PRAME(+)) have much more sensibility and specificity than HMB-45 in this specific diagnostic set [53,54,55,56].

##### Melan A/MART-1

Melan A/MART-1 is a melanoma-associated antigen that is recognized by autologous cytotoxic T lymphocytes, encoded by the *MLANA* gene located on chromosome 9p24.1, and involved in the formation and trafficking of melanosomes (specifically, of the melanocyte protein PMEL17, which is the antigen recognized by HMB45 and involved in the formation of stage II melanosomes) [57]. Melan A (18 kd) is a single-domain transmembrane that is exclusively expressed in melanocytes of the skin and retina (as proved by the analysis of messenger RNA in normal tissues) and intracellularly detectable in melanosomes and endoplasmic reticulum [57]. At present, several anti-MART-1 antibodies are commercially available, but those most commonly used in routine practice and for research purposes are the mouse monoclonal antibody clones M2-7C10 and A103 (used in our laboratory) [58]. Similar to HMB-45, MART-1 also shows a lower sensibility, compared to S-100 and SOX-10 (85–97% in primary cM, 57–86% in metastatic cM, and only 0–7% in desmoplastic cM), and it is negative in the majority of tumors that could be immune-histologically be confounded with cM; as result, in our daily practice routine, we often use MART-1 alone and/or in combination with HMB-45 (and with S-100 and SOX-10) in the above-mentioned diagnostic settings [51,52,58,59]. MART-1 strongly reacts with the junctional, intraepidermal, and dermal melanocytes in both cN and cM; we always performed it in conjunction with HMB-45, in order to evaluate the silhouette of the melanocytic lesion (symmetry/asymmetry), estimate the depth of invasion in cM, and assess the lympho-vascular invasion, adnexal involvement, and peri-adnexal extension [48,51,52,58]. However, dermatopathologists should be aware that: (1) cN with neurotization and/or stromal metaplasia, congenital cN, and hyper-maturating cN could completely lose or show a gradual diminishing of the expression of MART-1; (2) MART-1 could be expressed by adrenal cortical tumors, PEComa and related tumors, mesotheliomas, salivary gland tumors, and sex cord-stromal tumors (interestingly, some authors showed that these tumors do not produce MART-1 RNA, thus concluding that this “apparently paradoxical” positivity is related to an immunologically cross-reaction with unrelated antigens) [47,52,58,60,61,62]. Because of its high sensitivity for melanocytic lesions, MART-1 is a useful marker for the pathological evaluation of SLNB, in order to identify, but not to differentiate, NN and MM (both MART-1(+)) [62,63]. Besides, MART-1 has the advantage (over S-100 and HMB-45) of not being expressed in histiocytes and dendritic cells; as result, it is frequently used in association with the other immunohistochemical markers for the evaluation of SLNB [63,64].

#### 2.2.2. Markers Useful for the Differential Diagnosis between cN and cM

##### Ki67

Ki67 is a protein associated with cell proliferation, and it is encoded by the *MKI67* gene located on chromosome 10q26.2 [65]. It is expressed during all active phases of the cell cycle (late G1, S, G2, and mitosis, but not in G0 and early G1), and it is a reliable tool for evaluating the growth fraction of a cell population [65]. In recent studies, multiple distinctive molecular functions of Ki67 have been clarified, with cellular distribution and roles, depending on the cell cycle phase: distribution of heterochromatin antigens and nucleolar association of heterochromatin (interphase), formation of the perichromosomal layer, and prevention of mitotic chromosomes aggregations (mitosis) [66]. At present, the antibody adopted in the vast majority of laboratories (as well as in ours) do detect Ki67 as the mouse monoclonal antibody, clone MIB1 (it is often used as a synonym of Ki67, sometimes creating linguistic confusion) [67,68]. Several authors proved that Ki67 shows significant differences between cN and cM [51,52,65,66,67,68]. Specifically, conventional, Spitz, congenital, blue, and dysplastic cN exhibit positivity in about 1–3% of cells, usually disposed of at the dermal–epidermal junction with no/scattered positive cells in the deep part of the lesion (“dermal hot-spot” with Ki67 < 5%) [65,66,67,68,69,70,71,72]. By contrast, cM shows a higher percentage of positive cells (>15%), as well as a different staining pattern, with clustered positive cells in the deeper part of the lesion (“dermal hot-spot” with Ki67 > 5%) and/or a random pattern of staining [65,66,67,68,69,70,71,72]. Although, in the 2018 WHO Classification of Skin Tumors, Ki67 is strongly recommended for the differential diagnosis between dysplastic cN (<5%) and superficial spreading cM (>30%); in our personal experience, it is quite impossible to find “early” superficial spreading cM (those that raise more diagnostic problems with dysplastic cN) with such high Ki67 [7,8]. Besides, the pathologists should be aware of several diagnostic pitfalls in the application of Ki67 for the diagnosis of melanocytic neoplasms, namely cN with a high Ki67 index (recurrent/persistent cN, traumatized cN, proliferative nodules in congenital cN, etc.), cM that could display a Ki67 similar to that of cN (especially nevoid cM), and cN for which it is difficult to evaluate Ki67 only in the melanocytic component (cN with a high inflammatory component as halo cN, Meyerson cN, regressed cN) [65,66,67,68,69,70,71,72]. To reduce these pitfalls, several authors elaborated on “combined scoring systems” (integrating Ki67 with other markers to obtain a predictive score) and/or DS (Section 2.2.4), which allow for evaluating Ki67 only in the melanocytic component [73,74]. In our laboratory, we adopted DS (MART-1/Ki67 and HMB-45/Ki67) and found that more than the absolute value of Ki67 should be taken into account: (1) unusual, deep, and/or asymmetrical staining pattern of the dermal component; (2) Ki67(+) deep dermal cells with pleomorphism atypical nuclei; (3) Ki67(+) intraepithelial cells exhibiting pagetoid spreading (*personal observation, data unpublished*).

##### p16, p21 and p53

p16/INK4a (p16), p21/WAF-1 (p21), and p53 are all proteins involved in the regulation of the cell cycle and encoded by *CDKN2A*, *CDKN1A*, and *TP53* genes, which are located on chromosomes 9p21.3, 6p21.2, and 17p13.1, respectively [75,76]. p16, p21, and p53 are involved in the kinase-based signaling network known as the DNA damage response (DDR), which is responsible for the coordination of DNA damage cell cycle checkpoint activation and processes of DNA reparation [75,76]. Specifically, p16 and p21 belong to the CIP/KIP family of kinase inhibitors and play a critical role in cell cycle progression and senescence, mainly cooperating with Rb (“p16/Rb pathway”) and p53 (“p53/p21 pathway”); p53 is a master regulator of the cell cycle, apoptosis, and genomic stability, through several mechanisms (activation of DNA repair proteins, arrest of the cell cycle at the G1/S, initiation of the apoptosis, and senescence response to short telomeres) [75,76]. Several antibody clones (E6H4, JC8, and G175-405) have been developed for the detection of p16, but the most commonly used clone in surgical pathology laboratories (and in our laboratory) is the mouse monoclonal E6H4 [51,52,77,78]. p16 attracted great interest in the field of melanocytic pathology, since it has been shown that the biallelic/homozygotic inactivation of *CDKN2A* gene (and its corresponding loss of immunohistochemical expression) is a molecular step that is able to distinguish cM (p16 (-)) from cN (p16(+)) [77,78,79,80]. Numerous studies showed that cN stained (61–100%) p16 with a typical “mosaic/puzzle” staining; by contrast, only 12–80% of cM were p16(+) [77,78,79,80]. Nevertheless, the major limits of these studies were the criterion used to define p16 positivity (nuclear, cytoplasmatic, or both, as well as the percentage of positivity and pattern of staining) and differences between the cohorts (different histotypes of cN and cM, different stages of cM, primary vs. metastatic cM, etc.) [77,78,79,80,81,82,83]. It is well-known that specific histotypes of cN and cM preferentially show loss of p16, due to the relevance of the biallelic inactivation of *CDKN2A* for their oncogenesis process [81]. Besides, *CDKN2A* biallelic inactivation is recognized to be a late molecular step in the oncogenesis of cM, which is mainly involved in the advanced/metastatic stages (the percentage of metastatic cM p16(+) ranges between 0% and 41%); as result, p16 is not useful for the differential diagnosis of superficial lesions (superficial spreading cM and dysplastic cN), which represent the majority of routine diagnostic dilemmas [82,83]. In our experience, and in line with most of the literature data, the diagnostic scenarios in which p16 is mainly useful are the following: (1) the evaluation of dermal and/or nodular atypical melanocytic lesions/melanocytomas (the atypical Spitz tumor, atypical cellular blue tumor, and atypical proliferative nodule arising in congenital cN), where p16 loss reflects the biallelic inactivation of *CDKN2A* (also proved by molecular techniques) and represents strong criterion of malignancy; (2) the identification of a more aggressive phenotype is acquired by the primary cM, as p16 loss is characteristic of the advanced/metastatic cM; (3) differential diagnosis between NN and MM in the evaluation of SLNB [28,29,53,54,55,56,77,78,79,80,81,82,83,84]. Although some studies showed that PRAME is superior to p16 for discriminating NN from MM, in our experience, p16 remains a reliable diagnostic tool in this diagnostic setting [53,54,55,56,83,84]. Interestingly, we found very exceptional cases of cM that show a “paradoxical” diffusion and/or clonal overexpression of p16, representing a potential diagnostic pitfall and reflecting complex cell cycle deregulation, which results in the intracellular accumulation of p16 protein [85]. p21 protein exhibits an opposite pattern, compared to p16, with over-expression observed in cM and no- or hypo-expression in cN [86,87]. However, this molecule and its underlying molecular mechanisms are less known, compared to p16, and the immunohistochemistry for p21 is not frequently adopted in routine practice but mainly for research purposes [86,87]. At present, we use p21 (clone 4D10, mouse monoclonal) in our daily routine as an additional diagnostic tool, but only in selected scenarios for which the literature data are more substantial, such as Spitz (especially in acral sites) and mucosal melanocytic lesions [86,87,88,89,90]. Although the alterations of the *TP53* pathway are very frequent in cM, from a molecular point of view, these could underlie numerous genetic, epigenetic, and post-translational alterations, whose effects on the protein production (and, therefore, on our capability to immunohistochemically detect it) are very complex to predict [6,51,52,91,92]. Furthermore, the alterations of the *TP53* pathway are a late event in the carcinogenesis of cM (therefore, not so useful in the routine practice for the diagnosis of the most problematic cases) and could rarely also affect cN and melanocytic lesions with unpredictable biological potential [6,51,52,78,85,91,92,93]. At present, we use p53 (clone DO-7, mouse monoclonal) in our daily routine as an additional diagnostic tool, only in the context of desmoplastic melanoma, especially for the differential diagnosis between neurofibroma-like desmoplastic cM and neurofibroma [94].

##### PRAME

PRAME (preferentially expressed antigen in melanoma) is a tumor-associated antigen identified through the T-cell clones obtained from a patient with metastatic CM and encoded by the *PRAME* gene located on chromosome 21q11.22 [95]. PRAME is expressed in normal tissues (mainly male germline cells) and several tumors, with a large variety of functions in oncogenesis, immune response, apoptosis, and metastases [96,97,98]. Curiously, PRAME could act as an oncogene or tumor suppressor gene in different cancer types, exerting its biological functions through the regulation of its downstream targets (p53, p21, Bcl-2, TRAIL, RAR, Hsp27, and S100A4); besides, PRAME has a pivot role in the immunotherapy response and may be an attractive target for immunotherapy [96,97,98,99]. It became of great interest in the field of melanocytic tumors, as it proved to be expressed (and so immunohistochemically detectable) in cM but not in cN, potentially being the marker able to solve one of the most problematic issues of the surgical pathology [100]. Over the past years, several antibodies against PRAME have been developed, but the most commonly used antibody in routine practice, as well as the evaluation of melanocytic tumors, is the rabbit monoclonal, clone EPR20330 [100]. Lezcano et al. developed a score based on tumor cells with nuclear stain (0: 0%, 1+: 1–25%, 2+: 26–50%, 3+: 51–75%, 4+: ≥ 76%) and showed that it has a high sensibility and specificity in distinguishing cM and cN (4+: 87% of metastatic cM, 83.2% of primary cM, 93.8% of in situ cM, 94.4% of acral cM, 92.5% of superficial spreading cM, 90% of nodular cM, 88.6% of lentigo maligna melanomas, 35% of desmoplastic cM and only one case of Spitz cN; 0–1%: 86.4% of all cN, 100% of NN, and 100% of solar lentigo) [100]. The same authors found a 90% concordance between the PRAME score and cytogenetic tests results, supporting this marker as an important ancillary test (cheaper and faster, but not completely interchangeable with cytogenetic tests) for the diagnosis of complex melanocytic lesions [101]. Subsequently, other authors tested this antibody in the most problematic areas of the melanocytic pathology (atypical Spitz tumors, pauci-cellular lentigo maligna, nevus-associated cM, resections margins of lentigo maligna, NN and MM, etc.) and found very promising results; however, they adopted different cut-offs and raised the problem to correctly identify the exact percentage of the positive cells able to differentiate cN from cM as well as whether different percentages need to be adopted for different melanocytic lesions [102,103,104]. Besides, these results need to be validated in a large case series, with long-term follow-up, that is able to prove the real nature of ambiguous melanocytic tumors. Many other aspects have to be clarified before the adoption of this marker as the “answer to all our problems” (how to interpret “intermediate” results? How to interpret PRAME results in cases of a discordant molecular test?). Besides, PRAME is expressed in many other tumors (germ cell tumors of the testis, lymphomas, peripheral nerve sheath tumors, ovarian carcinomas, etc.) but not in the majority of desmoplastic cM (one of the most challenging melanocytic lesions), and we already suggested great caution before the adoption of PRAME as a “pan-melanoma” marker [96,97,98,99,100,105,106,107]. We recommend using PRAME in conjunction with DS, adopting a melanocytic marker (HMB-45 or MART-1) only in appropriately selected diagnostic settings and integrating this result with the histologic exam, other immunohistochemical analyses, and molecular techniques in “*really-difficult-to-diagnose*” melanocytic lesions. In our practice, we adopt this marker as an adjunctive diagnostic tool, especially for (1) ambiguous melanocytic lesions (atypical Spitz tumors vs. Spitz cM, high-grade dysplastic cN vs. early cM in situ, etc.); (2) differential diagnosis between NN and MM in selected difficult cases; (3) more accurate evaluation of surgical resection margins in lentigo maligna; (4) the distinction between the dermal “nevoid” component of nevoid cM and dermal cN in nevus-associated cM.

#### 2.2.3. Markers Useful for the Identification of Specific Histological Subtypes of cN and cM (BRAF V600E, c-Kit/CD117, ROS1, ALK, pan-TRK, BAP-1, β-Catenin, PRKAR1A, NF1, and IDH1)

Over the past years, the growing research in the field of molecular biology made it possible to identify the specific clinical–pathological entities that are characterized by specific molecular alterations, and the 2018 WHO classification of melanocytic lesions is mainly based on their molecular background and correlation with the entity of UV damage [6]. As result, the search of these genetic alterations has become fundamental for identifying and characterizing these new histological entities, thus allowing for a more detailed diagnosis and prognostic–therapeutic stratification (many of these molecular alterations identify potentially targetable therapeutic targets) [6]. The roles of these molecules depend on the analyzed molecules and could affect several aspects of the biology of melanocytes, cN, and cM (tumor growth, response to immunotherapy, apoptosis, etc.), but this is beyond the scope of this article [6]. Since these genetic alterations lead to an over- and/or aberrant expression of specific molecules, and the latter are associated with well-defined histological features of the melanocytic lesion, an expert dermatopathologist could suspect a specific genetic alteration from just the H&E exam and prove it with the immunohistochemistry [6,108,109,110,111,112,113,114,115,116,117,118,119,120,121]. In our routine practice, we do not use standard panels; the choice of immunohistochemical panels is performed case-by-case, based on the H&E exam. Specifically, the antibodies we use in our laboratory and specific histological entities, related to their over- and/or aberrant expression, are the following: -BRAF V600E: melanocytic lesions in intermittently sun-exposed skin (superficial spreading cM, simple lentigo, conventional and/or lentiginous cN, and dysplastic cN), deep-penetrating cN, *BAP1*-inactivated melanocytic lesions, pigmented epithelioid melanocytoma (PEM), acral melanocytic lesions (especially cM), nodular cM (less frequent), and nevoid cM (less frequent); -c-Kit/CD117: acral melanocytic lesions (especially cM), and lentigo maligna; -ALK, ROS1, pan-TRK (NTRK1, NTRK2, and NTRK3), RET, and MET: Spitz lesions (also Reed cN) and acral melanocytic lesions (especially cM); -β-catenin: deep-penetrating cN, rare cases of cM with a “deep-penetrating like silhouette”; -PRKAR1A: PEM; -BAP-1: *BAP1*-inactivated melanocytic lesions, cM arising in blue cN and atypical cellular blue tumor (rare cases); -NF1: lentigo maligna, desmoplastic cM, and acral melanocytic lesions (especially cM); -IDH1: recently introduced category of melanocytoma. In this diagnostic setting, although the immunohistochemistry has shown excellent concordance rates with molecular biology techniques (PCR, FISH, and NGS) and, therefore, represents a reliable, feasible, time- and money-saving tool for investigating these lesions, it is usually adopted as a “screening tool”, and the molecular biology techniques are adopted to confirm and expand the genetic background of the analyzed lesions [108,109,110,111,112,113,114,115,116,117,118,119,120,121]. These techniques, differently from the immunohistochemistry, are able to identify the specific mutations and/or fusion partners; these data could provide relevant pathogenetic, prognostic, and therapeutic indications (specific mutations are associated with pharmacological resistances; additionally, specific fusions are associated with marked pharmacological sensitivity to kinase inhibitors) [108,109,110,111,112,113,114,115,116,117,118,119,120,121]. A summary of these immunohistochemical markers and the histological entities related to their over- and/or aberrant expression is presented in Table 2; illustrative examples of some of these immunohistochemical markers and the corresponding histological entities are shown in Figure 2.

#### 2.2.4. Double Stains (DS) (HMB-45/Ki67, MART-1/Ki67, CD34/SOX10, HMB-45/PRAME, and MART-1/PRAME)

Over the past years, the development and application of DS have greatly increased in surgical pathology, due to the more detailed assessment of specific histopathological features (compared to the respective single stains), as well as saving of time, money, and histological material [122]. Specifically, in the field of melanocytic pathology, the most commonly used DS are those combining Ki67 with cytoplasmic melanocytic markers (HMB-45 and MART-1), thus allowing us to more correctly assess the proliferation index in only the melanocytes (ignoring the lymphocytes, keratinocytes, and endothelial cells) [73,74]. In our experience, these DS (HMB-45/Ki67 and MART-1/Ki67) are particularly useful in lesions that are almost exclusively junctional/intraepithelial, as well as in lesions with a high inflammatory infiltrate (halo cN, highly regressed cM, etc.). Other promising DS are those that allow us to correctly evaluate the presence of lympho-vascular invasion (D2-40/MITF, D2-40/SOX10, D2-40/S-100, and D2-40/MART-1), even if the obtained results and superiority are partially discordant compared to single stains and H&E stains and H&E [123,124,125]. In line with these previous works, we recently showed that DS for CD34/SOX10 (“pan-vascular marker” and “pan-melanocytic” marker) and H&E do not greatly differ in the lympho-vascular invasion detection rates and profiles of association with unfavorable pathological features of cM, arguing against the routinary adoption of this tool and suggesting that its adoption should be restricted to specific subgroups of cM [126,127,128]. As previously clarified (Chapter 2.2.2.), our working group has recently developed two DS combining PRAME (nuclear) with HMB-45 and MART-1 (cytoplasmatic) that showed very encouraging results and became part of the immunohistochemical panels routinely used in our laboratory [56,107]. In our experience, these DS (HMB-45/PRAME and MART-1/PRAME) are particularly useful in the following diagnostic scenarios: (1) lesions almost exclusively junctional/intraepithelial (allowing us to not evaluate keratinocytes); (2) lesions with a high inflammatory infiltrate (allowing us to not evaluate lymphocytes); (3) differential diagnosis between NN and MM, especially in SLNB; (4) metastasis of unknown primary tumor and/or primary cutaneous tumor with undifferentiated morphology, especially with limited available histological material.

## 3. Conclusions

Here, we summarize the current concepts and advances on the application of immunohistochemistry in the diagnosis of cN and cM. Despite continuous progress in the genetic classification of melanocytic lesions, there is still a need for improvements in the correct immunohistochemical characterization and diagnosis of this deadly disease. Hopefully, this diagnostic progress could result in an improvement regarding the therapeutic choices, as well as the reduction of mortality and morbidity by cM.

## Figures and Tables

**Figure 1 ijms-23-05911-f001:**
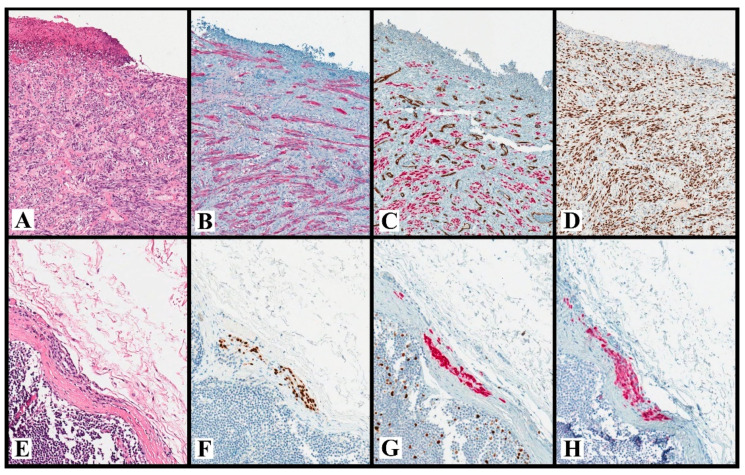
**Desmoplastic cM** (**A**–**D**). A case of ulcerated desmoplastic cM with marked desmoplasia, atypical spindle cells, and rare mitoses ((**A**): H&E, original magnification ×100). This case turns out positive for SOX10 ((**B**): CD34/SOX10, original magnification ×100; CD34: brown, SOX10: red), S100 ((**C**): S100, original magnification ×100), and p53 ((**D**): p53, original magnification ×100). Note that DS CD34/SOX100 shows the absence of lympho-vascular invasion (**B**), without SOX10(+) cells inside the vessels (labeled with CD34). **NN** (**E**–**H**). A small intra-capsular NN that histologically resembles cN, with bland nuclei and absence of mitoses ((**E**): H&E, original magnification ×200). This NN is positive for SOX10 ((**F**): SOX10, original magnification ×200), MART-1 ((**G**): MART-1/Ki67, original magnification ×200; MART-1: red, Ki67: brown) and p16 ((**H**): p16, original magnification ×200). Note that DS MART-1/Ki67 shows the absence of proliferating melanocytic cells (**G**), without MART-1(+)/Ki67(+) cells; by contrast, it shows proliferating lymphocytes MART-1(−)/Ki67(+) within the lymphoid follicles. Abbreviations: cM: cutaneous melanoma; DS: double staining; NN: nodal nevus.

**Figure 2 ijms-23-05911-f002:**
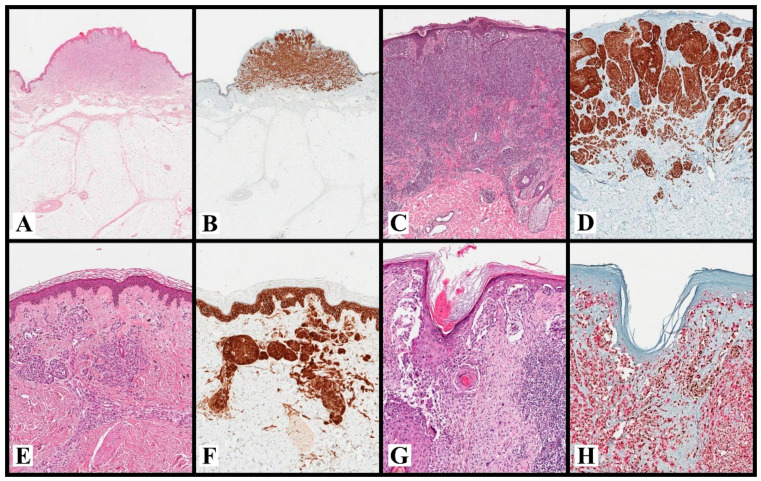
**Spitz cN with ALK-translocation** (**A**,**B**). A case of Spitz cN with a characteristic plexiform morphology ((**A**): H&E, original magnification ×20), which suggests an ALK translocation and turned out positive for ALK ((**B**): H&E, original magnification ×20). **Spitz cN with NTRK1-translocation** (**C**,**D**). A case of Spitz cN with filigree-like rete ridges and lobulated nests ((**C**): H&E, original magnification ×50), which suggests a NTRK1 translocation and turned out positive for NTRK ((**D**): H&E, original magnification ×50). **Deep-penetrating cN with CTNNB1 mutation** (**E**,**F**). A dermal cN with a wedge-shaped silhouette, large and bland epithelioid melanocytes arranged in small nests, and ill-defined fascicles ((**E**): H&E, original magnification ×80), which suggests a “deep-penetrating” morphology and turned out positive (cytoplasmatic and nuclear) for β-catenin ((**F**): H&E, original magnification ×80). **Lentigo maligna melanoma** (**G**,**H**). A lentigo maligna melanoma with single and nested atypical melanocytes that involve the adnexal structures and markedly efface the epidermis ((**G**): H&E, original magnification ×100). This case turns out positive for HMB-45 (cytoplasmatic) and PRAME (nuclear), with a 4+ score ((**H**): HMB-45/PRAME, original magnification ×100; HMB-45: red, PRAME: brown). Note that DS HMB-45/PRAME allows us to simultaneously establish the melanocytic nature of the lesion and evaluate the PRAME score. Abbreviations: cN: cutaneous nevus; DS: double staining.

## Data Availability

Not applicable.

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
