# Peer review of "Cutaneous Melanomas: A Single Center Experience on the Usage of Immunohistochemistry Applied for the Diagnosis"

_ijms, 2022, doi:10.3390/ijms23115911_

Round 1
Reviewer 1 Report
In the manuscript "Cutaneous Melanomas: Current Concepts and Advances in Immunohistochemistry applied for the Diagnosis" by Costantino Ricci et al., the authors attempt to perform a comprehensive review about the use of different imunohistochemistry markers in the diagnosis of different lesions in the spectrum of cutaneous melanoma. The theme is interesting and extremely relevant for the field. In addition, it is a broad topic, which poses an enormous challenge for a non-biased summary of relevant information. Unfortunately, the current manuscript title is misleading, since the authors present a very personal and biased perspective of their experience with the use of different markers instead of performing a literature-based review on the theme. Thus, the title should be reformulated to something like: "Cutaneous Melanomas: a single center experience on the usage of immunohistochemistry applied for the diagnosis". In addition, the paper is written in a confusing manner and it is not well-organized. Furthermore, some claims are not clearly supported by references in the literature.
The specific aspects that should be improved and merit the attention of the authors are the following:
- Some parts of the text are too superficial to be informative and relevant for the reader. For example, it is the case of the small introduction section, the histological exam section and the section on markers useful for the identification of specific histological subtypes.
- Table 1 also contains information that is questionable and innacurate. For example, regarding SOX-10, besides the correct estimation of intraepithelial pagetoid spreading, it can also be used to study the spread on lentiginous melanocytic proliferations and to determine the depth of invasion of melanocytic lesions (to estimate the Breslow depth).
- Regarding point 6 of SOX-10 in Table 1, the usage of SOX-10 in the differential diagnosis between scar and desmoplastic cM (especially in the excisional enlargement of desmoplastic cM) is a controversial topic in the literature, with some studies advising against its usage. Please discuss this topic and mention some recent papers like:
- Emily L Behrens et al. SOX-10 staining in dermal scars. Journal of Cutaneous Pathology, 2019.
- Matthew R Donaldson and L Arthur Weber. SOX10 commonly stains scar in Mohs sections. Dermatology Online Journal, 2020.
- HMB-45 is used in the evaluation of a gradient of melanocyte maturation in melanocytic lesions. Such information is missing in table 1.
- MELAN-A/MART1 is used in the identification of the lympho-vascular invasion, the adnexal involvement and the peri-adnexal extension in cM, but we can also use SOX-10 for this. This information should be added to the SOX-10 section in Table 1.
- Table 1 mentions that PRAME is used in the evaluation of melanocytic lesion, however, such sentence is not informative. In fact, how does PRAME expression really help in this context? The sentence needs to be more clear.
- In the hands of the authors, the identification of the lympho-vascular invasion in cM can be performed using a double staining CD34/SOX-10. However, is this the only method? Can use use CD31/SOX-10 and Podoplanin/SOX-10?
- When describing the SOX-10 marker the authors should make clear that SOX-10 is a marker expressed during embryonic development and that later marks all cells derived from the neural crest.
- The section 2.2.3 on the markers useful for the identification of specific histological subtypes could be considerably improved if a table could be included containing the selected markers and the specific lesions where they are expressed/mutated with detailed informations. In this section, the authors should also discuss the current role of PCR-based techniques and NGS studies for the classification of melanocytic tumours.
Therefore, the present review paper attempts to summarize relevant infomation about immunohistochemistry markers in the diagnosis of cutaneous melanona, but it will be mandatory to perform extensive changes in the present manuscript if it is considered relevant for publication in the International Journal of Molecular Sciences.
Author Response
We are grateful to Reviewer 1 for these stimulating comments and observations, which allowed us to ameliorate our paper. We revised our manuscript following your appreciated suggestions and we provided our point-by-point list of responses. We thank again Reviewer 1, and we hope that the revised manuscript can be accepted for this Journal.
We agree with Reviewer 1 that in this review the literature data are integrated with our personal experience (as also remarked in the abstract, lines 24-26), and therefore we modify the Title as suggested by you ("Cutaneous Melanomas: a single center experience on the usage of immunohistochemistry applied for the diagnosis").
- Some parts of the text are too superficial to be informative and relevant for the reader. For example, it is the case of the small introduction section, the histological exam section and the section on markers useful for the identification of specific histological subtypes.
Thank you so much to Reviewer 1 for these observations, which give us the possibility to make the Manuscript more explicative and useful for the readers. As for "Introduction" and "Histological exam", it is difficult to make these parts truly informative for a Manuscript like this (without being redundant and/or off topic) but we needed them to present the topic and introduce the reader to the core of the Manuscript (besides, the "format" of the article has been given to us by the Editor Staff). However, we tried to slightly modify and make them more informative and suitable for the Manuscript. Regarding your comments on "markers useful for the identification of specific histological subtypes", we really appreciate them and we modify the article accordingly (please read the answer to comment number 9). - Table 1 also contains information that is questionable and innacurate. For example, regarding SOX-10, besides the correct estimation of intraepithelial pagetoid spreading, it can also be used to study the spread on lentiginous melanocytic proliferations and to determine the depth of invasion of melanocytic lesions (to estimate the Breslow depth).
Thank you so much to Reviewer 1 for this observation, which gives us the possibility to modify and make Table 1 more explicative and useful for the readers. We modified Table 1 accordingly to this and the other (please see below) observations regarding Table 1. - Regarding point 6 of SOX-10 in Table 1, the usage of SOX-10 in the differential diagnosis between scar and desmoplastic cM (especially in the excisional enlargement of desmoplastic cM) is a controversial topic in the literature, with some studies advising against its usage. Please discuss this topic and mention some recent papers like:
- Emily L Behrens et al. SOX-10 staining in dermal scars. Journal of Cutaneous Pathology, 2019.
- Matthew R Donaldson and L Arthur Weber. SOX10 commonly stains scar in Mohs sections. Dermatology Online Journal, 2020.
Thank you so much to Reviewer 1 for this observation, which gives us the possibility to discuss a really crucial topic. We know that there could be many diagnostic pitfalls to adopting SOX10 to this diagnostic field (positive histiocytes, frozen section in Mohs surgery, entrapped nerves, fibroblasts with "neuroid" features and many others); however, we believe that SOX10, always according to H&E exam, could be really helpful in this diagnostic set. We modified the Manuscript in line with your suggestion, discussed this topic and changed all the subsequent References. - HMB-45 is used in the evaluation of a gradient of melanocyte maturation in melanocytic lesions. Such information is missing in table 1.
Thank you so much to Reviewer 1 for this observation, which gives us the possibility to modify and make Table 1 more explicative and useful for the readers. We modified Table 1 accordingly to this and the other (please see below) observations regarding Table 1. - MELAN-A/MART1 is used in the identification of the lympho-vascular invasion, the adnexal involvement and the peri-adnexal extension in cM, but we can also use SOX-10 for this. This information should be added to the SOX-10 section in Table 1.
Thank you so much to Reviewer 1 for this observation, which gives us the possibility to modify and make Table 1 more explicative and useful for the readers. We modified Table 1 accordingly to this and the other (please see below) observations regarding Table 1. - Table 1 mentions that PRAME is used in the evaluation of melanocytic lesion, however, such sentence is not informative. In fact, how does PRAME expression really help in this context? The sentence needs to be more clear.
Thank you so much to Reviewer 1 for this observation, which gives us the possibility to modify and make Table 1 more explicative and useful for the readers. We modified Table 1 accordingly to this and the other (please see below) observations regarding Table 1. - In the hands of the authors, the identification of the lympho-vascular invasion in cM can be performed using a double staining CD34/SOX-10. However, is this the only method? Can use use CD31/SOX-10 and Podoplanin/SOX-10?
Thank you so much to Reviewer 1 for this observation, which gives us the possibility to clarify a crucial point. We report in "Immunohistochemistry" and "Double stains" chapters all the other double stains potentially useful and previously tested for the identification of lympho-vascular invasion in cM (D2-40/MITF, D2-40/SOX10, D2-40/S-100, D2-40/MART-1). We also reported our experience with a double stain developed by our laboratory (CD34/SOX10) and clearly specified that "according to the previous publications evaluating D2-40/MITF, D2-40/SOX10, D2-40/S-100, D2-40/MART-1, the results obtained with these double stains do not greatly differ with those of H&E, so arguing against the routinary adoption of these tools and suggesting as their adoption should be restricted to specific subgroups of cM". - When describing the SOX-10 marker the authors should make clear that SOX-10 is a marker expressed during embryonic development and that later marks all cells derived from the neural crest.
Thank you so much to Reviewer 1 for this observation, which gives us the possibility to clarify a crucial point. We modified the Manuscript and we added/specified this crucial point. - The section 2.2.3 on the markers useful for the identification of specific histological subtypes could be considerably improved if a table could be included containing the selected markers and the specific lesions where they are expressed/mutated with detailed informations. In this section, the authors should also discuss the current role of PCR-based techniques and NGS studies for the classification of melanocytic tumours.
Thank you so much to Reviewer 1 for this observation, which gives us the possibility to improve our Manuscript and expand a relevant topic. We added Table 2 with all this information and briefly discussed the role of PCR and NGS in this scenario (unfortunately this topic is very complex and probably requires a specific and dedicated article, and we do not want to be off-topic).
Reviewer 2 Report
This review, “Cutaneous Melanomas: Current Concepts and Advances in Immunohistochemistry applied for the Diagnosis” aims to present all the data related to the immunohistochemistry of cM, discussing its application for diagnosis, prognostic characterization and treatment of this deadly disease, also briefly summarizing the role of these molecules in the biology of melanocytes and cM. The scientific work is very interesting, I believe that the manuscript is potentially appropriate for publication in this journal.
Author Response
We are grateful to Reviewer 2 for these beautiful comments, and we really appreciate them. We thank again Reviewer 2, and we hope that the revised manuscript can be accepted for this Journal.
Round 2
Reviewer 1 Report
In the revised version of the original manuscript "Cutaneous Melanomas: Current Concepts and Advances in Immunohistochemistry applied for the Diagnosis" by Costantino Ricci et al., the authors attempted to perform a comprehensive review about the use of different imunohistochemistry markers in the diagnosis of different lesions in the spectrum of cutaneous melanoma. The theme is interesting and extremely relevant for the field. In addition, it is a broad topic, which poses an enormous challenge for a non-biased summary of relevant information. The authors present a personal and literature-based perspective of their experience with the use of different markers in the diagnosis of cutaneous melanoma. The authors addressed in a satisfactory manner all my previous concerns regarding the paper and, thus, I have no further objections towards its publication in the International Journal of Molecular Sciences.